# Modeling Movement Disorders via Generation of hiPSC-Derived Motor Neurons

**DOI:** 10.3390/cells11233796

**Published:** 2022-11-27

**Authors:** Masuma Akter, Baojin Ding

**Affiliations:** Department of Biochemistry and Molecular Biology, Louisiana State University Health Sciences Center, Shreveport, LA 71130-3932, USA

**Keywords:** hiPSC, motor neurons, small molecules, transcription factors, movement disorders

## Abstract

Generation of motor neurons (MNs) from human-induced pluripotent stem cells (hiPSCs) overcomes the limited access to human brain tissues and provides an unprecedent approach for modeling MN-related diseases. In this review, we discuss the recent progression in understanding the regulatory mechanisms of MN differentiation and their applications in the generation of MNs from hiPSCs, with a particular focus on two approaches: induction by small molecules and induction by lentiviral delivery of transcription factors. At each induction stage, different culture media and supplements, typical growth conditions and cellular morphology, and specific markers for validation of cell identity and quality control are specifically discussed. Both approaches can generate functional MNs. Currently, the major challenges in modeling neurological diseases using iPSC-derived neurons are: obtaining neurons with high purity and yield; long-term neuron culture to reach full maturation; and how to culture neurons more physiologically to maximize relevance to in vivo conditions.

## 1. Introduction

Movement disorders are a group of neurological conditions that cause either increased or decreased or slow movements. The movements may be voluntary or involuntary, and implicated in many neurological diseases, such as Dystonia, amyotrophic lateral sclerosis (ALS), Ataxia, Parkinson’s disease [1], and so on. Motor neurons (MNs) are a remarkably powerful cell type in the central nervous system (CNS), and they are involved in both autonomic and voluntary movements. Most prior research of movement disorders was carried out using patient postmortem tissues or rodent models [2,3]. However, some subtle alterations in brain tissues and the dysregulations in specific neuronal subtypes may be transient and therefore difficult to preserve and detect in posthumous patient tissues [4]. While animal models provide insights into disease mechanisms, significant species-dependent differences exist, and animal models only mirror the limited aspects of the pathophysiology of human diseases. It is believed that these species-dependent differences caused the high failure rate in clinical trials that have been derived from successful results in animal models [5,6,7]. Additionally, when using posthumous patient tissues or rodent models, it is difficult to decipher the molecular pathogenesis via biochemical approaches, which require a large number of high-purity living neurons. Human induced pluripotent stem cells (hiPSCs)-derived neurons overcome the limited access to human brain tissues and provide an unprecedented approach to model human neurological diseases [8].

iPSC-based disease modeling starts from Yamanaka and his colleagues’ groundbreaking studies, which demonstrated that somatic cells could be reprogrammed into pluripotent stem cells by ectopic expression of four transcription factors (Oct4 (O), Sox2 (S), Klf4 (K), and c-Myc (M)) under defined culture conditions [9,10]. An essential requirement for fulfilling the potential of hiPSCs is the ability to reliably differentiate into all three germ layers (ectoderm, mesoderm, and endoderm) and generate specific cell types with defined phenotypic traits [11,12,13]. The development of iPSCs offered a new approach for patient specific stem cell-based research, bypassing the reliance on overexpression models, interspecies differences of animal models, and also ethical concerns of using human embryonic stem cells (hESCs) [14]. Given that iPSCs derived from somatic cells can preserve the donor’s genetic background, less transplant rejection will occur when they are used for cell therapy. The in vitro phenotypes of disease-specific iPSC-derived cells hold the most promise to bridge the gap between the clinical phenotype and fundamental molecular and cellular mechanisms, creating new strategies for drug screening and novel therapeutic interventions [15,16]. Additionally, CRISPR engineering of iPSCs enables researchers to make paired patient mutation lines and isogenic control lines, greatly facilitating the research for understanding disease pathogenesis [12,13,17,18,19].

It is also well known that iPSCs and iPSC-derived cells have limitations, especially in the context of culture heterogeneity and dosage variability. The best route of administration and survivability in the hostile inflammatory microenvironment is controversial [20]. Improved ways of making cells, gene-editing technologies, along with patient-derived iPSC cells, have revolutionized the generation of experimental disease models. This provides an unlimited supply of any type of cells from once-inaccessible human tissues for research [21,22]. Neurodevelopmental and neurodegenerative disorders are particularly approachable using iPSC technology as iPSC-derived neurons retain the patient’s genomic context and provide an excellent cellular model system in deciphering the pathophysiology of diseases. Modulating the combination, concentration, and exposure time of crucial signaling molecules has yielded protocols for generating neurons and glia from iPSCs [23]. Thus far, researchers have developed protocols by which several neural and glial cell types could be generated, including glutamatergic neurons [24], GABAergic neurons [25,26], dopaminergic neurons [27], serotonergic neurons [28], MN [29,30,31,32], astrocytes [33], and microglia [34,35]. The improvement techniques and updated protocols in neural differentiation and maturation greatly facilitate the generation of iPSC-derived cells.

On the other hand, iPSC-based disease modelling is relatively new in biomedical research. Substantial progress has been made in developing differentiation protocols for the generation of different cell types. Excitingly, some in vitro models can recapitulate aspects of neuronal synaptic networks, which greatly advances functional modeling of neurodevelopmental and neurodegenerative diseases. Recent studies generating iPSC-derived MNs from movement disorders of ALS [36,37,38] and DYT1 patients [8,31,39] showed great potential to experimentally model molecular events underlying disease pathogenesis. The generation of patient-specific MNs provides an unprecedented approach in modeling MN-related disorders and deciphering cellular and molecular pathogenesis. In this review, we focused on the techniques for the generation of hiPSC-derived cholinergic MNs.

## 2. Generation of hiPSC-Derived MNs

Many studies are attempting to generate mature MNs from hiPSCs for modeling movement disorders. Developmental biologists have identified the signaling molecules and transcription factors that are involved in MN differentiation and maturation, providing the foundation for the generation of hiPSC-MNs. Currently, scientists are using these extrinsic factors to guide the MN differentiation from stem cells, thereby mimicking regionalization processes during nervous system development. Many protocols have been developed that rely upon core signaling pathways, which could synchronize neuronal induction to MN-specific signaling cascades and upregulate the expression of MN-specific genes.

The generation of hiPSC-MNs is a consecutive process that consists of a variety of induction stages, including iPSC induction, embryoid body (EB) formation, neural rosette growth, neuron progenitor cells (NPCs) differentiation, and MN induction and maturation (Figure 1A). At each induction stage, different culture media with different supplements will be employed to support and drive the cell fate towards MNs. At each differentiation stage, the cells show different growth patterns with unique cellular morphology and express specific protein markers (Figure 1B). These features can be used to verify the cell identity at each induction stage and perform quality controls to obtain highly pure MNs. In the following sections, we will discuss the process of iPSC-MN generation with a focus on the approaches using small molecules (chemicals) and lentiviral delivery of transcription factors. Meanwhile, the regulatory mechanisms of MN differentiation at different stages will be discussed.

### 2.1. Generation of NPCs from hiPSCs

Human nervous system development begins with neural induction converting ectodermal cells into neuroectodermal cells, leading to the formation of the neural plate and finally the neural tube [40,41]. How neural regionalization is precisely controlled to form the forebrain, midbrain, hindbrain, and spinal cord along with the rostral-caudal (R-C) axis remains unknown. Dorso-ventral, rostral, and caudal patterning are very important events in neural tube maturation and specification. The gradients of various morphogens such as the canonical WNT/β-catenin signaling pathway (Wnt), retinoic acid (RA), and Sonic Hedgehog (Shh) signaling are involved in the determination of neuroaxis formation [42,43,44]. Insights from the model animals show that the rostral-caudal (R-C) axis patterning of NPCs is controlled by modulating the Wnt and RA signaling, whereas dorso-ventral patterning is controlled by the modulation of Shh signaling [45,46,47,48]. Most neuronal differentiation schemes mimic embryonic developmental signals by small molecule patterning. This initial study further showed that different combinations of small molecules used as patterning factors could push NPCs toward distinct neuronal fates [49,50,51]. The neuroectoderm is specified by inhibition of mesoderm and endoderm differentiation factors and acquires an initial rostral neural character through the regulation of BMP (bone morphogenetic protein), TGFβ (transforming growth factor-β), FGF (fibroblast growth factor), and Wnt signaling [52,53,54,55,56,57]. The neural fate can be potentially induced by the inhibition of BMP and TGFβ. These rostral neural progenitors are caudalized in response to Wnts, FGFs, and RA during early development [4].

One of the many ways of generating different types of neurons is through the differentiation of NPCs, which are a homogenous, multipotent, undifferentiated, self-renewable cell population. NPCs are committed to become neural lineage and to be differentiated to specific neuronal types in defined culture conditions [58]. iPSCs are induced to become NPCs through neuralization, either by EB formation [59,60,61,62], or by dual SMAD inhibition in monolayers [49,63,64]. EB formation is the most widely used method for neuralization [65,66,67,68,69]. In this process, cells grow in suspension and spontaneously arrange in three-dimensional aggregates upon withdrawal of factors promoting pluripotency. The Dual-SMAD-Inhibition method was developed in 2009 by Chambers and colleagues [49]. This method used a small molecule (Noggin, SB431542) to suppress the TGF-β/Activin/Nodal pathway and the BMP-canonical pathway [70,71]. After a 5-day exposure of the hiPSCs to Noggin/SB431542, the cells became an early-stage neuroepithelial population with expression of SOX1, Paired box protein 6 (PAX6), and Zic family member 1 (ZIC1) markers, and were able to form neural rosette organization [49,72,73]. This protocol showed an 80% efficiency of hESC and hiPSC differentiation into PAX6-positive NPCs. Once the iPSCs cells differentiate into NPCs, the newly formed NPCs continue expressing neuroepithelial markers, such as SOX1 (SRY-Box Transcription Factor 1), SOX3 (SRY-Box Transcription Factor 3), (PSA-NCAM) Polysialylated-neural cell adhesion molecule, and MS1 (Musashi RNA Binding Protein 1) [74,75,76]. Reinhardt and colleagues reported a protocol that NPCs can be generated using only small molecules. Neural induction was introduced through inhibition of both BMP and TGF-β signaling using Dorsomorphin and SB43152. To stimulate the canonical Wnt signaling, CHIR99021, a GSK3β inhibitor, was added to the cell medium, and the Shh pathway was stimulated by using purmorphamine. These neural progenitors are also able to differentiate into different types of neurons, including MNs [70].

Recently, two approaches were reported for generation of NPCs using a combination of small molecules: either RA and VPA (Valproic Acid) or LDN-193189 (SMAD inhibitor) and SB431542 (Activin/BMP/TGF-beta Pathway Inhibitor) [29,77,78]. Both combinations can successfully generate NPCs, which can be spontaneously differentiated into neurons consisting of glutamatergic (~75%), GABAergic (~15%), and dopaminergic (TH+) (~5%) neurons under defined culture conditions. These NPCs can also be differentiated into highly pure (90%) cholinergic MNs via transduction of three transcription factors: NEUROG2 (Neurogenin 2), ISL1 (LIM homeobox 1), and LHX3 (LIM homeobox protein 3) [29,31,32].

### 2.2. MN Induction via Small Molecules

Small molecules are bioactive compounds that can modulate specific cellular pathways involved in cell signaling, transcription, metabolism, or epigenetics, all of which are modulated during cellular reprogramming. If selective epigenetic modulation can be achieved with chemicals, it could remodel the chromatin structure and activate the gene expression of transcription factors, achieving similar effects to the ectopic expression of reprogramming factors. The advantage of using small molecules in reprogramming is that their biological effects are typically rapid, reversible, and dose-dependent, allowing precise control over specific outcomes by fine-tuning their concentrations and combinations. In Table 1, we summarized the roles of small molecules, including chemicals and peptide growth factors, in MN differentiation. These small molecules target different signaling pathways (Table 2) that promote MN differentiation and maturation. In Table 3, we summarized the chemical cocktails that have been reported in recent publications for the generation of iPSC-MNs.

To acquire the MN progenitor identity, NPCs need to caudalize and ventralize with the action of RA and Shh, respectively [42,109]. Shh induces the expression of homeodomain transcription factors and basic helix-loop-helix (bHLH) [11], the critical intermediaries in the control of cell patterning and neuronal cell fate determination [110,111,112,113]. The combined actions of RA and Shh are thought to establish a spatial and temporal regulation of the expression of transcription factors, such as LHX1 (homeodomain transcription factors) [114,115,116], OLIGO2 (oligodendrocyte transcription factor), MNX1 (MN and pancreas homeobox 1, HB9), and ISL1 [42]. All of these factors are necessary for the subsequent differentiation of NPCs to MNs [117,118,119]. Several studies have identified that OLIG2 is a bHLH protein, which is essential for establishing MN progenitor identity downstream of Shh signaling. OLIG2 also has a key role in specifying the subtype identity and pan-neuronal properties of developing MNs [117,120,121,122]. The expression of downstream transcriptional regulators, particularly HB9, a homeodomain protein expressed in postmitotic MNs, is crucial to consolidate MN identity [123,124,125].

Small molecules have a profound influence on neural induction and promote hiPSC differentiation into MNs combined with the simultaneous inhibition of TGFβ activin, Nodal, and BMP (also known as dual SMAD inhibition) signaling. Dual SMAD inhibition is a well-established method which utilizes small molecules to block endodermal and mesodermal cell fates and promote neuroectoderm conversion. It dramatically enriches neural ectoderm from pluripotent cells with a high expression of PAX6 and sex-determining region Y-box 2 (SOX2). Noggin and SB431542 are the most commonly used dual SMAD inhibitory small molecules. Noggin acts as a BMP inhibitor and SB431542 inhibits the Lefty/Activin/TGFβ pathways by blocking the phosphorylation of ALK4, ALK5, and ALK7 receptors. The GSK-3β inhibitor promotes neural progenitor proliferation by stimulating the canonical Wnt signaling pathway, which contributes to the maintenance of neural precursors. Molecular activation of SHH by Purmorphamine, canonical WNT signaling, and neural patterning by RA have been critical for MN induction. Previous studies have found that Shh induces upregulation of transcription factors OLIG2, NK2 homeobox 2 (NKX2.2), and neurogenin2 (NGN2) to direct the expression of MN fate-consolidating genes such as HB9 and ISL1. Neurotrophic and growth factors, such as BDNF, GDNF, and NT3 are also used as supplements to facilitate MN growth, maturation, and survival [8,29,126]. The entire differentiation process requires from 15 days to up to 2 months to generate fully functional MNs. In 2002, Wichterle and colleagues reported that RA and Shh were used to differentiate mouse ESCs into MNs through EB formation [127,128]. Wada and colleagues differentiated hESCs from the human and monkey ESCs toward MNs through neural rosette formation. They treated ESCs with 1 μM RA and 500 ng/mL Shh, leading to neural precursors becoming Tubulin β III+, Hb9+, Islet1+, and choline acetyltransferase-positive (ChAT+) neurons [129].

Scientists have been working continuously to advance our understanding of MN differentiation and to improve the techniques for generating MNs from iPSCs. The rapid advancement of RNA sequencing technologies contributed to a deep understanding of transcriptome composition and has discovered a large number of non-coding RNAs (ncRNAs) that participate in MN differentiation (Table 4). These ncRNAs have intense regulatory activities in a wide range of biological processes, including neuronal development, subtype diversification, specification, differentiation, and function [130,131,132]. Among the ncRNAs, long non-coding RNAs (lncRNAs) and miRNAs (microRNAs) are especially abundant in the nervous system and have been shown to be implicated in MN development and function.

Mature miRNAs are ∼22-nucleotide single-stranded RNAs that can recognize the 3′ untranslated region (UTR) of its target mRNAs and negatively regulate gene expression post-transcriptionally [146,147,148]. miRNAs are an integral part of the genetic program controlling MN survival and acquisition of subtype-specific properties [137,138,149]. Several studies also demonstrated that miRNAs mediated post-transcriptional regulation participates in fine-tuning the program of MN progenitor specification [148,150], MN differentiation [151,152,153], and subtype diversification [134,135,136,137,154,155]. miR-9 is involved in fine-tuning the differentiation of MN subtypes. Notably, a recent study revealed that miR-9 is transiently expressed during MN differentiation and regulates the expression of FoxP1 (Forkhead Box P1) and HOX (Homeobox (HOX) transcription factors. These transcription factors play a critical role in coordination of MN subtype identity and connectivity. In mice, overexpression of miR-9 induces neuronal differentiation by inhibiting the nuclear receptor [134,135], suggesting that miR-9 plays a role in fine-tuning the process of specification of MN subtype identity. Other studies showed that ISL1 expression by Onecut transcription factors (OC1) was important to generate LMC MNs [133,156,157]. Studies showed that miR-9 and OC1 are in mutually exclusive patterns in the embryonic spinal cord and miR-9 efficiently represses OC1 expression, demonstrating that regulation of OC1 by miR-9 is a crucial step in the specification of spinal MNs.

miR-218 is the most abundant and highly enriched miRNA in developing and maturing MNs [136,137,158]. miR-218 is decreased in human ALS postmortem spinal cord, and cell-free miR-218 can serve as a marker for MN loss in a rodent model of ALS [159,160]. Studies showed that in the developing spinal cord, the expression of miR-218 is directly upregulated by the Isl1–Lhx3 complex, which drives MN fate. Inhibition of miR-218 suppresses the generation of MNs in both chick neural tube and mouse ESCs, suggesting that miR-218 plays a crucial role in MN differentiation [137]. Previously, it has been found that complete loss of miR-218 results in the breakdown of neuromuscular synaptogenesis, hyperexcitability, post-natal lethality, MN loss, and complete paralysis [136,154,155,161].

Chen and colleagues demonstrated that the repression of Olig2 mRNA (MN progenitor marker) is controlled by mir-17-3p microRNA [138,150]. The expression of miR-17-3p is repressed by Shh, which results in elevated expression of Olig2. Thus, a high amount of Shh will direct neuronal progenitors to differentiate toward MNs [162,163]. Functional studies indicate that miRNA plays a significant role in a broad range of cellular and developmental processes of subsets of MN. By using an in vitro model of human spinal MN development, it has been shown that miR-375 is strongly activated during spinal motor neurogenesis and its expression is specific to MNs [130]. Knockdown of miR-375 significantly impairs MN differentiation, highlighting its essential role in MN development. miR-375 also protects MNs from DNA damage-induced degeneration by inhibiting p53 and therefore preventing apoptosis. Downregulation of the miR-375-3p in patients with spinal muscular atrophy leads to an increase of the p53 protein level and thus to apoptosis [144,145].

Long non-coding RNAs (lncRNAs) are RNAs that exceed 200 nucleotides in length, and they are not translated into proteins. LncRNAs participate in various stages during MNs differentiation, including guiding neural fate choice by driving transcription factor localization [164,165,166], regulating local translation at synapses [167,168], influencing MN development, and contributing to the pathogenic mechanisms underlying MN diseases (MNDs) [164,169].

The lncRNA CAT7 (chromatin-associated transcript 7) is a polyadenylated lncRNA that lies upstream (~400 kb) of MNX1 (MN homobox1). CAT7 has been found to temporally regulate MNX1 expression during the early stages of human ESC-MN differentiation. Loss of CAT7 causes de-repression of MNX1 before committing to motor neuronal lineage [170]. Another lncRNA Meg3 plays a critical role in maintaining postmitotic MN cell fate by repressing progenitor genes that regulate the differentiation of MN identity [171,172,173].

NEAT1 (nuclear-enriched abundant transcript 1) is a well-characterized lncRNA that functions as a chromatin regulator and organizes nuclear structures called ‘paraspeckles’. Paraspeckles contain proteins involved in transcription and RNA processing [174,175]. NEAT1 is highly enriched in neurons of the anterior horn of the spinal cord and in the cortical tissues of ALS patients. Nishimoto and colleagues demonstrated NEAT1 upregulation and increased paraspeckle formation in the MNs during the early phases of ALS pathogenesis [176,177,178]. The exact role of NEAT1 still needs to be resolved. Given their dynamic expression patterns in MNs and emerging roles in MN development and function, it is not surprising that dysregulation of noncoding RNAs has been implicated in MNDs. Understanding the mechanisms of action and functions of lncRNAs may assist the development of new therapies for MNDs.

### 2.3. MN Induction via Lentiviral Delivery of Transcription Factors

Previous reports have underscored the essential roles of transcription factors in MN development (Table 5). Decades of developmental studies have identified key signaling molecules and cell-intrinsic transcriptional programs [179,180] that specify MN identity during embryonic development. These transcription factors were transduced into neural progenitor cells after differentiation from ESCs/iPSCs, and MNs can be obtained 11 days after the transduction. These transcription factors are generally introduced into the cells via viral transduction (generally the most efficient ones) [29,181]. This transcription factors-mediated differentiation has been shown to produce highly efficient functional MNs with repetitively firing MNs after two weeks post-viral infection (wpi) [31,32].

Transcription factors ISL1 and LHX3 are sufficient to activate a MN gene expression program in other neural progenitors and increase post-mitotic specification by directly reprogramming pluripotent stem cells into MNs [188,189,190]. Specifically, a combinatorial expression of LIM homeodomain transcription factors Lhx3 and Isl-1, together with the expression of the pro-neural gene NGN2, have been shown to be critical to induce MN specification during development [114,191,192,193]. The combinatorial expression of Lhx3 and Isl1 will form an Isl1-Lhx3-hexamer, which will trigger MN specification in a chick spinal cord, ESCs, and iPSCs [114,194,195,196,197]. The binding of the Isl1-Lhx3 complex activates the transcription of genes that are essential for MN specification such as HB9 and promotes the expression of a wide range of terminal differentiation genes, including a battery of cholinergic pathway genes that enable cholinergic neurotransmission [198,199].

The combination of different transcription factors was often used to obtain a high quality of iPSC-MNs. These transcription factors need to be delivered into cells via lentiviral vectors or other vehicles (Table 6). In 2013, the group of Hynek Wichterle demonstrated that overexpression of three transcription factors (Ngn2, Isl1 and Lhx3) was sufficient to rapidly and efficiently program spinal MN identity from the mouse ESCs. Replacement of Lhx3 by Phox2a (Paired Like Homeobox 2A) led to the specification of cranial, rather than spinal MNs. Isl1-Lhx3 and Isl1-Phox2a heterodimers showed different DNA-sequence preferences for the basis of cell reprogramming, indicating that there are synergistic interactions between programming factors underying specification of alternate MN fates [189,200,201,202]. Goto et al. have used a single sendai virus-mediated overexpression of the TF cocktail NGN2, ISL1 and LHX3 in both mice and human iPSCs to promote the expression of MN markers. Notably, after 3 weeks of differentiation, NGN2/ISL1/LHX3-overexpressing neurons were electrophysiologically active and formed neuromuscular junctions (NMJ) with cultured myocytes [203]. The MNs derived via this method from ALS patient’s iPSCs have also shown disease phenotypes. De Santis et al. expressed transcription factors of Ngn2, Isl1, and Phox2a in human iPSCs via Piggy-bac transposable vectors and converted human iPSCs into cranial MNs and upregulated pan-MN genes such as TUBB3, ISL1, and ChAT within the first 3 days of differentiation. HB9 expression was increased when LHX3 was co-expressed, whereas PHOX2B and TBX20 (T-Box Transcription Factor 20) were detected by day 5 [144]. Finally, the authors of this study functionally characterized the cranial MNs obtained after 12 to 13 days to observe that these cells were capable of firing action potentials upon current stimulation, and almost half of all analyzed cells even displayed spontaneous glutamatergic postsynaptic currents.

Goparaju et al. showed that overexpression of NGN1 (Neurogenin 1), NGN2 (Neurogenin 2), NGN3 (Neurogenin 3), NEUROD1 (Neuronal Differentiation 1), and NEUROD2 (Neuronal Differentiation) in human PSCs combined with RA, forskolin, and dual SMAD inhibition via SB431542 and dorsomorphin yields highly pure neuronal cultures expressing the MN markers HB9, ISL1, and ChAT [103]. MNs generated from this method have shown functional activity (repetitive action potentials and calcium-transient) within a week. We have reported that a single lentiviral vector expressing three factors (NGN2, ISL1, and LHX3) is necessary and sufficient to induce iPSC-derived MNs (iPSC-MNs) [29,32]. MNs derived using these methods robustly expressed general neuron markers, such as microtubule-associated protein 2 (MAP2), neurofilament protein (SMI-32), tubulin β-3 class III (TUBB3), and MN-specific markers HB9 and CHAT. These MNs showed electrical maturation within 3 weeks [32].

## 3. Quality Control: Validation of Neuron Identity and Purity

To ensure that the high quality and purity of MNs can be obtained from hiPSCs, it is necessary to validate the cell identity at each stage during the process of induction and differentiation. Specific markers at different stages could be examined and used to estimate the induction quality and the MN purity (Figure 1B). As the passage number of iPSC may affect the differentiation of iPSC-derived neurons [207], using a lower passage number is recommended in the generation of hiPSC-derived MNs.

### 3.1. Markers of Early Induction from hiPSC to NPC

Patient-specific somatic cells, such as skin fibroblast cells and peripheral blood mononuclear cells (PBMC), can be reprogrammed into the pluripotent state through ectopic expression of Yamanaka factors (Oct3/4, Sox2, Klf4, and c-Myc). The iPSC must be fully characterized to ensure quality before differentiation into neurons or other cell types. The morphology of iPSCs should demonstrate a typical hESC-like appearance composed of tightly packed cells in phase contrast microscopy (Figure 1B). A healthy iPSC line should robustly express the pluripotency markers (SSEA4, TRA-1-81, OCT3/4, and SOX2) and show a normal karyotype [12,13]. There are several techniques that can be used to assess the pluripotency. The pluripotency markers could be examined at protein levels by immunocytochemistry (ICC) and western blot or by using real-time PCR to measure transcription levels. Newly generated iPSC lines need to be validated that they can differentiate into three germ layers: endodermal, mesodermal, and ectodermal cell lineages. This can be verified by two approaches: 1) in vitro differentiation of iPSCs in suspension to form EBs, which highly express trilineage markers [59,208], and 2) EBs injected into immunocompromised NOD/SCID mice will form teratomas, which consist of three germ layers [209,210]. Similarly, three germ layers specific markers can be examined using ICC or western blot at protein level or RT-PCR at transcription level. The iPSCs and differentiated cells need to be confirmed as mycoplasma-negative before establishing a cell line stock.

### 3.2. Markers of MNs at Early Immature Stages

To achieve more accurate disease modeling and maximize the potential applications, quality controls are critical to verify cell identity and purity. Many methods could be used, including specific marker expression, molecular and functional properties, cellular morphology assay, electrophysiological analysis, and animal transplantation. Many protocols analyze samples within 2 to 5 weeks from the onset of differentiation. Early stages of MNs showed the typical polygonal cell body with few and short dendrites (Weeks 1–2). However, the generation of ISL1/2 and HB9-expressing cells can vary from 3 days to 2 weeks after differentiation. Longer differentiation protocols have an advantage over shorter ones when preparing fully functional mature neurons, which could be used to examine the electrophysiological activity and synaptic network. After treatment with lentivirus-expressing-specific transcription factors or small molecules, usually within a week the cell will become neuron-like with condensed nuclei, long axons, and multiple neurites. Generation of MNs requires the identification of genes that are expressed at the initial stages of MN differentiation. At day 13 of differentiation, studies observed the expression of early MN-specific factors, PAX6, OLIG2, ISL-1 and NEUROD [191]. PAX6 and OLIG2 are required to initiate a general MN fate differentiation [43,117,119]. Studies have also found that ISL1 is the earliest marker involved in the establishment of MN fate. The expression of MN-specific genes becomes evident at week 3 or later, including strong expression of LHX3, ISL1, and HB9. Early MNs are commonly characterized by transient co-expression of HB9 and ISL1/2 [123,211]. Homeobox gene HB9 is required for the consolidation of MN identity, and its expression is restricted to somatic MNs of the hypoglossal nucleus. Transcriptional upregulation of MN markers such as ISL1, HB9, and OLIG2 was shown in the early stage of MN differentiation in several studies.

### 3.3. Markers of MNs at Late Mature Stages

As HB9 is downregulated over the course of MN maturation, the expression of choline acetyltransferase (ChAT) appeared and temporally increased [212]. As ChAT is an enzyme responsible for the synthesis of neurotransmitter acetylcholine in cholinergic MNs; the expression of ChAT indicates that the cells reach maturation stage. miR-218, abundantly and selectively expressed in maturing MNs, is being recently used as a molecular marker to identify MNs [136,137].

The final maturation can be achieved with the action of neurotrophic factors (BDNF, GDNF, NT3) and demonstrated by increased dendrites arborization and cell-cell connections (weeks 2–6). Mature MNs display larger soma, increased cell shape, complexity of neurite outgrowth, and electrophysiological properties. iPSC-derived MNs are generally considered mature after 3 weeks of differentiation [100]. Higher expression of neuronal markers of TUBB3, MAP2, non-phosphorylated neurofilament heavy chain (SMI32), ChAT, and vesicular acetylcholine transporter (vAChT) indicates the maturation of the MNs. Monitoring the electrophysiological status of MNs in vitro is currently the most comprehensive method to assess their maturation. To demonstrate maturation, MNs are Synapsin-positive and electrophysiologically active. The electrophysiologically mature MNs are able to fire repetitive action potentials and generate spontaneous activity that requires the development of intrinsic (e.g., sufficiently polarized resting membrane potentials) and extrinsic (e.g., synapse formation) properties. Mature MNs are also capable of recreating NMJs when cocultured with myotubes in vitro and expressing acetylcholine receptor (AChR) clusters. All these characteristics indicate that hiPSCs had efficiently differentiated into fully functional MNs.

## 4. Modeling Neurological Diseases Using hiPSC-Derived MNs

Modeling MN-related diseases using hiPSC-based approaches requires culture conditions in a dish that can recapitulate the events underlying MN differentiation, maturation, aging, and degeneration. Several protocols exist to generate MNs from hiPSCs, and these cells have been used to study the pathophysiology of MN-related diseases, such as ALS, spinal muscular atrophy, and DYT1 dystonia [213,214,215,216,217,218]. The known disease-dependent cellular deficits are excellent features that can be used to validate the hiPSC-MNs in modeling the disease. For example, using patient-specific MNs, we have identified disease-dependent cellular deficits in DYT1 dystonia, including abnormal nuclear envelope morphology, disrupted neurodevelopment, impaired nucleocytoplasmic transport, and mislocalized nuclear Lamin B1 [31,39]. For each preparation of DYT1 iPSC-MNs, we routinely examined these cellular features to ensure that the materials we used are valid in modeling DYT1 dystonia. To identify and characterize functional molecular features of MNs, different technologies have been employed, ranging from fluorescence-based antibody staining of target markers to RNA-seq analysis. We usually perform ICC, Western blot, PCR, and qRT-PCR to verify the expression levels of neural markers at different stages during the induction process. Examination of well-characterized molecular markers is useful for determining the accuracy and efficiency of a protocol. Whole-transcriptome sequencing or RNA-seq, single cell sequencing is also often used to further characterize the expression of these markers in an unbiased manner. Modeling ALS with iPSC-derived MNs has recapitulated several known pathological findings in patient-derived cells, reinforcing the high value of the approach. Compared with healthy controls, patient iPSC-derived cells can be used to identify disease phenotypes at different levels, including molecular profiles, cellular features, and physiological functions. The development of high-throughput single-cell transcriptomics has changed the paradigm, empowering rapid isolation and profiling of MN nuclei, revealing remarkable transcriptional diversity within the skeletal and autonomic nervous systems. In addition, patient-specific iPSCs may also serve as powerful resources for personalized medicine, including drug discovery, genetic testing, and ultimately cell replacement therapy.

## 5. Future Challenges and Perspectives

Generation of hiPSC-MNs overcomes the limited access to human brain tissues and provides an unprecedented approach for modeling MN-related diseases, offering an excellent platform for developing therapeutic treatments. However, several challenges arise from using this cellular system for disease modeling [32]. First, the purity and yield of iPSC-derived neurons. Current human iPSC-MN induction protocols vary in timescale (ranging from 15 days to more than six weeks) and efficiency, with few protocols achieving both high efficiency and rapid MN generation [219,220]. Some studies require a large number of MNs with high purity, such as transcriptomic studies, to elucidate the alterations of genome-wide gene expression and proteomic studies to identify dysregulated factors in diseased neurons. Second, to obtain fully functional and mature neurons. Because of the lack of simplified and consistent protocols, the generated hiPSC-derived MNs in most studies are often functionally immature and heterogeneous. Some disease-dependent cellular and molecular deficits cannot be noticed until neurons reach full maturation, especially for age-related neurodegenerative diseases such as ALS, Alzheimer’s disease (AD), and Parkinson’s disease (PD). Modelling these late-onset diseases usually requires the long-term culture of neurons from several weeks to a few months, during which the neuron survival and potential contaminations are huge challenges. Third, how to culture the neurons more physiologically? Although chemical or physical modifications of the cell culture plates, such as coating with extracellular matrix (ECM), have been shown to be an efficient method to better mimic in vivo cell behavior [221], the outcomes of some experiments using in vitro cellular systems could be very different from the studies using in vivo models. iPSC-derived neurons cocultured with glial cells and the development of hiPSC to brain organoids under three-dimensional culture conditions could maximize the relevance to in vivo conditions.

To resolve these issues, studies are needed to understand the precise regulatory mechanisms of neural differentiation and maturation using in vivo models. Meanwhile, the protocols for the generation of iPSC-derived neurons need to be updated, simplified, and finally standardized to obtain consistent outcomes for biomedical research. The generation of different neuronal subtypes requires different protocols consisting of different induction factors, specific culture conditions and supplements, and particular treatments and processes. Generally, induction efficiency and neuron survival are paramount to achieve good yield and purity. The combinations of small molecules and/or transcription factors will enhance the induction efficiency and maximize the purity, while the optimized culture conditions will promote neuron maturation and survival, which will lead to a higher yield. One major difficulty in obtaining fully mature iPSC-MNs is the long-term culture issue, in which both neuron survival and potential contaminations are huge challenges. We generate iPSC-MNs using lentiviral delivery transcription factors, and these MNs can reach full maturation at 3 wpi with characterization of high expression of presynaptic proteins, cholinergic markers, and firing action potentials. For setting up experiments that require fully mature neurons, we usually culture iPSC-MN for 4 wpi [29]. For such a long-term culture, culture media supplemented with neurotrophic factors and neurons cocultured with astrocytes are required. For chemical-induced iPSC-MNs, once the MN identity is verified, in theory, the neurons will reach full maturation after culture for a long enough time. However, we have not directly compared the maturation, the culture time, and the detailed characterizations of iPSC-MNs that are generated by different approaches. It is hard to say which approach is better than the other. Excitingly, the techniques of genome-editing and three-dimensional culture of brain organoids greatly expanded the applications of hiPSC in disease modeling, cell therapy, and drug development [222,223].

## Figures and Tables

**Figure 1 cells-11-03796-f001:**
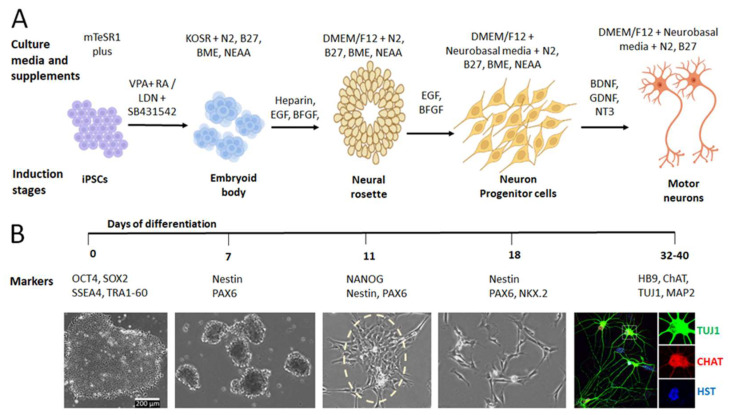
Generation of iPSC-derived MNs. (**A**) Schematic shows the process of generating iPSC-MNs. The culture media with different supplements are shown at different induction stages. (**B**) The timeline of the induction process from hiPSC to MNs. At each stage, specific markers and typical cellular morphology are shown. Image of MNs was adapted from [32].

**Table 1 cells-11-03796-t001:** Roles of small molecules in neuron differentiation.

Chemicals	Functions	References
Retinoic acid (RA)	Agonist for RA receptors. Promotes neural differentiation.	[14]
Valproic acid (VPA)	Histone deacetylase inhibitor. Facilitates the reprogramming of fibroblasts into iPSCs. Promotes neuronal differentiation.	[79]
SB431542	Inhibitor of TGF-β, Activin and Nodal signaling. Differentiation of human ES and iPSCs into neural progenitors. Increase in reprogramming efficiency in combination with other small molecules.	[49]
CHIR99021	Selective inhibitor of glycogen synthase kinase 3 (GSK-3). Enables reprogramming of fibroblasts into iPSCs. Induces neuronal differentiation.	[49]
Purmorphamine (PUR)	Sonic Hedgehog (Shh) activator. Improves the efficiency of MN differentiation.	[80]
Dorsomorphin	Inhibitor of both activin/nodal/TGF-*β* and BMP pathways. Induces rapid and high-efficiency neural conversion in both hESCs and hiPSCs. Induces neuronal differentiation in vitro.	[81]
Y-27632	Highly potent and selective inhibitor of Rho-associated, coiled-coil-containing protein kinase (ROCK). Improves embryoid body (EB) formation efficiency. Enhances survival of hESC during cell passaging.	[82]
Forskolin (FSK)	Stimulates adenylate cyclase activity and increases cAMP. Regulates neuronal specification and promotes axonal regeneration.	[83]
Compound E	NOTCH signaling inhibitor. Accelerates MN maturation.	[84]
Brain-derived neurotrophic factor (BDNF)	Activates TrkB signaling. BDNF enhances the survival and differentiation of neurons in vitro. Critical for neuronal survival, morphogenesis, and plasticity.	[85]
Glial cell line-derived neurotrophic factor (GDNF)	Activates tyrosine kinase receptor signaling. Promotes neuronal differentiation in later culture periods. Potential roles in various pathways, mediating growth, differentiation, and migration of neurons.	[86]
Ciliary neurotrophic factor (CNTF)	Neurotrophic factor. Promotes the survival of different neurons and the differentiation of neural progenitor cells (NPCs) in vitro.	[87]
Neurotrophin-3 (NT3)	Neurotrophic factor-mediated Trk receptor signaling. Neurotrophic factors promote the survival of neurons. Growth factor involved in stem cell differentiation.	[88]
Basic fibroblast growth factor (bFGF)	Fibroblast growth factor family. Stimulates hESC to form neural rosettes. Supports the maintenance of undifferentiated human hESCs.	[89]
Epidermal growth factor (EGF)	Mitogen. Induces the in vitro and in vivo proliferation of neural stem cells, their migration, and their differentiation towards the neuroglial cell line.	[90]
Heparin	Promotes the growth of hESCs. Supports the binding of FGF to its receptor and increases the stability of FGF. Activates Wnt signaling for neuronal morphogenesis.	[91]

**Table 2 cells-11-03796-t002:** Signaling pathways participating in MN differentiation.

Cell Signaling	Functions	References
Sonic Hedgehog (Shh) signaling	Shh signaling is required for the final specification of MNs. Activator: Purmorphamine (PUR); Inhibitors: Cyclopamine and HPI-1.	[92]
Dual SMAD inhibition	Block endodermal and mesodermal cell fates and promote neural conversion. Drive the rapid differentiation of hESCs and hiPSCs into a highly enriched population of NPCs. Inhibitors: SB431542, LDN193189, Noggin, LY364947, RepSox, Dorsomorphin, DMH-1	[93]
Neurotrophic factors signaling	Improves MN survival and maturation. Activators: BDNF, GDNF, NGF, NT-3	[94]
Wnt/β-catenin signaling	Contributes to patterning, proliferation, and differentiation throughout vertebrate neural development. Activator: CHIR99021 (CHIR), Kenpaullone, SB216763 Inhibitor: APCDD1, Waif1/5T4	[95]
Notch signaling	Regulates the balance between MN differentiation and the maintenance of the progenitor state. Activator: Dll, Dl4 Inhibitor: DAPT, LY411575	[96]

**Table 3 cells-11-03796-t003:** Chemical cocktails used for MN differentiation from iPSCs.

Chemical Cocktail and Cytokines	Target Signaling Pathways	Cellular Markers	Efficiency	Days	References
Chir-99021, SB431542, LDN1931899, RA, SAG, DAPT, BME, Ascorbic Acid, and Y-27632	Wnt, FGF, SHH signaling	OLIG2, ISL1/2, HB9, LHX1/2, FOXP2	~77%	14	[68,97]
RA, PUR, Y-27632, VPA and CHIR99021	SHH signaling, WNT/b-catenin	HB9, ISLET-1, ChAT	~85%	30–40	[98]
SB431542, CHIR99021, RA, PUR, BDNF, GDNF	SHH, WNT/b-catenin, and Notch	TUJ1, MAP2, HB9, ChAT, SYP	>85%	28	[99]
SB 431542, CHIR99021, dorsomorphin, and Cpd E	Activin/nodal/TGF-β and BMP pathways, SHH signaling	ChAT, HB9, SOX11, PAX6, nestin, OLIG2, TUJ1, MAP2	~88%	21	[100]
SB 431542, dorsomorphin, BDNF, RA, and ISL1/2	TGF-β, Activin, Nodal, and canonical	FOXP1, OXA5, MAP2, TUJ1	>40%	24	[101]
SB 431542, CHIR99021, dorsomorphin, and RA	Activin/nodal/TGF-β, BMP and GSK-3	ChAT, HB9, SMI-32	~80%	24	[102]
SB 431542, dorsomorphin, B18R, synTFs mRNAs of neurogenin and NeuroD families, FSK, BDNF, GDNF, and NT-3	Activin/nodal/TGF-β and BMP pathways	ChAT, HB9, and ISL1	~86%	12	[103]
RA, SAG, BDNF, GDNF, and DAPT	Neurotrophic factors, canonical signaling	ChAT, HB9, ISL1, SMI-32, TUJ1	70–95%	32	[104]
Dorsomorphin, SB431542, CHIR99021, RA, PUR, ascorbic acid, dibutyryl cAMP	Activin/nodal/TGF-β and BMP pathways	OLIG2, SOX2, ISLET1, AP2, HB9, SMI32, TUJ1	~70	45	[105]
SB431542, DMH1, CHIR99021, RA, PUR, Cpd E	BMP, Activin, WNT, SHH and NOTCH	NKX2.2, OLIG2, ISL1, MNX, TUJ1, ChAT, BTX	>90%	28	[84]
PUR, RA	Shh, Agonist for retinoic acid receptors	HB-9, TUJ1,OLIG2	>85%	28	[106]
Compound C, RA, cAMP Neurotrophic factor	BMP and Activin signaling	TUJ1, MAP2 Synapsin I, HB9, ChAT	~70%	20	[107]
PUR, RA	Shh signaling agonist	HB9, ISL1/2, ChAT, OLIG2	~80%	15	[108]

**Table 4 cells-11-03796-t004:** Noncoding RNAs and their functions in MN differentiation.

Noncoding RNA	Function	References
miR-9	miR-9 modifies MN columns by a tuning regulation of transcription factor FoxP1 (Forkhead box protein 1) levels in developing spinal cords.	[133,134,135]
miR-218	Expression of miR-218 is directly upregulated by the Isl1–Lhx3 complex, which drives MN fate. Inhibition of miR-218 suppresses the generation of MNs in both chick neural tube and mouse embryonic stem cells.	[136,137]
mir-17~92	Confers MN subtype survival during development.	[136,138,139]
mir-27	mir-27 as a major regulator coordinates the temporal delay and spatial boundary of Hox protein expression, which contributes to the specification of MN subtype identity.	[140]
miR-183-5p	miR-183-5p is a central regulator of MN survival under stress conditions. Increased miR-183-5p is correlated with cell stress in MNs of ALS in pre-symptomatic and early symptomatic stages.	[141,142]
miR-196	The timing and rostro-caudal extent of Hoxb8 activity in the neural tube is tightly regulated by miR-196, a miRNA species encoded within three Hox gene clusters. miR-196 effectively suppresses endogenous Hoxb8.	[143]
miR-375	miR-375 facilitates human spinal MN development and protects MNs from DNA damage-induced degeneration.	[130,144,145]

**Table 5 cells-11-03796-t005:** Roles of transcription factors in MN differentiation.

Transcription Factor	Functions	References
Neurogenin 2 (NEUROG2)	Transcriptional regulator and actively involved in neuronal differentiation. Unique and critical role in determining MN cell-type identity.	[182]
Sex determining region Y-box 2 (SOX2)	Critical for early embryogenesis and for maintaining embryonic stem cell pluripotency.	[183]
ISL LIM homeobox 1 (ISL1)	ISL1 is a major transcription factor necessary for MN identity. Fusion protein Isl1–Lhx3 specifies MN fate differentiation.	[184]
LIM homeobox 3 (LHX3)	Transcriptional activator involved in the development of interneurons and MNs.	[184]
POU class 5 homeobox 1 (POU5F1)	Critical for early embryogenesis and for embryonic stem cell pluripotency. Master regulator of initiation, maintenance, and differentiation of pluripotent cells.	[80]
Achaete-scute family basic helix-loop-helix transcription factor 1 (ASCL1)	Promotes cell cycle exit and develops neuronal progenitors and differentiation when expressed in neural progenitor cells.	[185]
POU Class 3 Homeobox 2 (POU3F2)	Plays potential role in morphological complexity, maturity, and action potentials of the neuronal cells	[186,187]

**Table 6 cells-11-03796-t006:** Transcription factors used in generation of iPSC-MNs.

Transcription Factors Delivered	Delivery Vector	Cellular Markers	Efficiency	Days to Reach Maturation	References
NGN2, ISL1, LHX3	Lentiviral	MAP2, SMI32, TUBB3, HB9, ChAT	>95%	35 days	[29,32]
NGN2	Lentiviral	ChAT, HB9, SMI-32, ISL1, FOXP1, MAP2, TUJ1	~95%	30 days	[204]
NGN2, SOX1, ISL1, and LHX3	Lentiviral	HB9, ChAT, TUBB3, MAP2, and synapsin	45–50%	35 days	[29,31]
POU5F1(OCT4) and LHX3	Lentiviral	MAP2, TUJ1, HB9, ChAT	70 ~ 90%	28 days	[205]
NGN2, ISL1, LHX3 and NGN2, ISL1, PHOX2A	Piggy-bac transposable	PHOX2B, TUJ1, ISL1, ChAT	~90%	11–12 days	[144,206]
NGN2, ISL1, LHX3	Sendai virus	HB9, MAP2, ChAT, Tuj1,	~93%	14 days	[203]
NGN2, ISL1, LHX3	Adenoviral	HB9, CHAT, SMI-31, HOXC6	60–70%	30 days	[191]
Ascl1, Brn2 (POU3F2), Myt1l, Hb9 (MNX1), NGN2, ISL1, and LHX3	Retroviral	MAP2, vChT, HB9, ISL1	~60%	35 days	[187]

## Data Availability

Not applicable.

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
