# Peer review of "Modeling Movement Disorders via Generation of hiPSC-Derived Motor Neurons"

_cells, 2022, doi:10.3390/cells11233796_

Round 1
Reviewer 1 Report
I would appreciate the authors for the review article 'Modeling movement disorders via generation of iPSC-derived motor neurons. The authors have compiled updated information about the different methods to differentiate iPSCs in motor neurons and written it in a well-organized manner. So far no method can generate fully mature motor neurons, author should also include how mature MNs can be obtained in different methods and which is the most efficient tool.
Author Response
Re: Thanks for the positive evaluation about this manuscript.
The major barrier to obtain fully mature motor neurons is the long-term culture issue. In my lab, we use lentiviral delivery transcription factors method, and iPSC-MN can reach full maturation at 3 weeks post-viral infection (wpi), characterized by high expression of presynaptic proteins, cholinergic markers, and firing action potentials. We usually culture iPSC-MN at 4 wpi for setting up experiments that require fully mature neurons. This long-term culture requires optimized culture condition, including proper supplements and coculture with astrocytes. For chemical induced MNs, once the MN identity is verified, culture neurons for a longer time, in theory, the neurons will reach the full maturation. However, we have not directly compared the maturation, the culture time, and the detailed characterizations of iPSC-MNs that are generated by different approaches. It is hard to say which approach is better than another one. We added this information in the revised manuscript in the section of Future Challenges and Perspectives.
Reviewer 2 Report
I enjoyed reading and reviewing this well-presented review of the subject on “Modeling movement disorders via generation of hiPSC-derived 2 motor neurons”, which would be a useful addition to the literature. The language is generally good, but the whole manuscript needs to be reviewed for fine grammar and punctuation.
The abstract in its current form seems incomplete. Further information could be added with the approach of making it conclusive. For instance which methodology seems to be the preferred choice for differentiating iPSCs to neurons or what problems scientific community is facing and what reader can anticipate before reading the article.
-As you mentioned in the sub-section 3.1, about assessment of the quality of iPSCs; the success of neuronal differentiation depends on the quality. very recently some articles suggested that iPSCs passage numbers also play role in assuring the neuronal differentiation quality, even the iPSC markers remain unchanged. https://www.nature.com/articles/s41598-022-19018-6
Probably you can discuss this in this review article as well.
-The future challenges section could be further improved. It can be more structured in terms of a new hypothesis that can be executed to utilize the hiPSCs-technology in the field.
Author Response
English language and style are fine/minor spell check required. I enjoyed reading and reviewing this well-presented review of the subject on “Modeling movement disorders via generation of hiPSC-derived 2 motor neurons”, which would be a useful addition to the literature. The language is generally good, but the whole manuscript needs to be reviewed for fine grammar and punctuation.
Re: Thanks for the positive evaluation about this manuscript. We have double checked the language and corrected the grammar and punctuation errors in the revised manuscript.
The abstract in its current form seems incomplete. Further information could be added with the approach of making it conclusive. For instance which methodology seems to be the preferred choice for differentiating iPSCs to neurons or what problems scientific community is facing and what reader can anticipate before reading the article.
Re: Thanks for the suggestion. We updated the abstract with addition more information as suggested.
As you mentioned in the sub-section 3.1, about assessment of the quality of iPSCs; the success of neuronal differentiation depends on the quality. very recently some articles suggested that iPSCs passage numbers also play role in assuring the neuronal differentiation quality, even the iPSC markers remain unchanged. https://www.nature.com/articles/s41598-022-19018-6 Probably you can discuss this in this review article as well.
Re: Thanks for the suggestion. In Section 3.1 of the revised manuscript, we briefly discussed this interesting point about the effects of iPSC passage numbers on the differentiation of neurons and cited this paper.
-The future challenges section could be further improved. It can be more structured in terms of a new hypothesis that can be executed to utilize the hiPSCs-technology in the field.
Re: Thanks for the suggestion. We updated this section with addition of more perspectives based on our understanding and changes, and the subtitle was changed to Future Challenges and Perspectives.